# A Pharmacokinetic Study of Mix-160 by LC-MS/MS: Oral Bioavailability of a Dosage Form of Citroflavonoids Mixture

**DOI:** 10.3390/molecules27020391

**Published:** 2022-01-08

**Authors:** Jesús Alfredo Araujo-León, Rolffy Ortiz-Andrade, Efrén Hernández-Baltazar, Emanuel Hernández-Núñez, Julio César Rivera-Leyva, Víctor Yáñez-Pérez, Priscila Vazquez-Garcia, Carla Georgina Cicero-Sarmiento, Juan Carlos Sánchez-Salgado, Maira Rubí Segura-Campos

**Affiliations:** 1Laboratorio de Cromatografía, Facultad de Química, Universidad Autónoma de Yucatán, Merida 97069, Mexico; jalfredoaraujo@gmail.com (J.A.A.-L.); priscila.vgarcia@hotmail.com (P.V.-G.); 2Laboratorio de Farmacología, Facultad de Química, Universidad Autónoma de Yucatán, Merida 97069, Mexico; cicero_carla@comunidad.unam.mx; 3Laboratorio de Tecnología Farmacéutica, Facultad de Farmacia, Universidad Autónoma del Estado de Morelos, Cuernavaca 62209, Mexico; efrenhb@uaem.mx; 4Departamento de Recursos del Mar, Centro de Investigación y de Estudios Avanzados del Instituto Politécnico Nacional-Unidad Mérida, Merida 97205, Mexico; emanuel.hernandez@cinvestav.mx; 5Laboratorio 4, Facultad de Farmacia, Universidad Autónoma del Estado de Morelos, Cuernavaca 62209, Mexico; julio.rivera@uaem.mx; 6Bioterio de la Escuela de Medicina, Universidad Anáhuac-Mayab, Merida 97302, Mexico; victor.yanez@anahuacmayab.mx; 7Programa de Maestría y Doctorado en Ciencias Médicas, Odontológicas y de la Salud, Facultad de Medicina, Universidad Nacional Autónoma de México, Ciudad Universitaria, Ciudad de Mexico 04510, Mexico; 8Hypermedic MX, Ciudad de Mexico 04930, Mexico; juanc.sanchez@live.com; 9Facultad de Ingeniería Química, Universidad Autónoma de Yucatán, Merida 97203, Mexico; maira.segura@correo.uady.mx

**Keywords:** LC-MS/MS, flavonoids, hesperidin, naringenin, pharmacokinetics

## Abstract

This study was performed to evaluate and compare the pharmacokinetic parameters between two dosage formulations of hesperidin and naringenin: mixture and tablet. Our objective was to determine that the flavonoid tablet does not significantly modify the pharmacokinetic parameters compared with the mixture. For this study, we administered 161 mg/kg of either mixture (Mix-160) or tablet composed of hesperidin and by intragastric administration. Blood microsamples were collected from tail vein up to 24 h. Serum flavonoid extraction was performed by solid phase extraction and analyzed by LC-MS/MS of triple quadrupole (QqQ). Serum concentration vs. time plot showed that data fitted for a first-order model. The pharmacokinetic parameters were calculated by a noncompartmental model. The results showed that the absorption constant is higher than the elimination constant. The first concentration was found at five minutes, and minimal concentration at 24 h after administration, suggesting a enterohepatic recirculation phenomena and regulation of liver cytochromes’ activity. We did not find meaningful differences between the pharmacokinetic parameters of both samples. We concluded that tablet form did not interfere with the bioavailability of hesperidin and naringenin, and it could be a suitable candidate for developing a drug product.

## 1. Introduction

Diabetes and hypertension are chronic degenerative diseases with a high prevalence worldwide. There is remarkable interest because they are risk factors for developing metabolic syndrome [1]. In recent years, there has been a growing interest in the research of bioactive molecules with multitarget affinity for treating different health conditions at once [2]. This approach should reduce polypharmacy of current treatments [3].

Our research group is focused on investigating natural products-based molecules, mainly citroflavonoids. Naringenin and hesperidin are good examples of molecules with multitarget affinity. Previous reports have shown that the mixture of these flavonoids produced vasorelaxant activity in rat aorta rings and antihypertensive effects on animal models [4]. Additionally, toxicological assays in animal models showed that both molecules are safe because they did not exert severe toxicological effects. This evidence suggests that they could be classified as low-risk molecular entities, useful substances for drug development. According to results, a mixture of naringenin/hesperidin could be a good candidate for developing a dosage form with antidiabetic and antihypertensive properties [5].

In the pharmaceutical market, there are flavonoid-based medicines, such as Daflon^®^ (Laboratorios Sanfer S.A. de C.V., Mexico), which is a mix of two flavonoids, diosmin and hesperidin, indicated for chronic venous insufficiency [6], and Fabroven^®^ (Pierre Fabré Pharma, France), which is a formulation based in Ruscus aculeatus extract (flavonoids-rich root extract), hesperidin methyl chalcone, and ascorbic acid for rheological blood disorders, such as lymphatic insufficiency [7].

Based on the above information, we decided to develop a tablet with the same concentration of Mix-160 as a solid dosage form prototype. Then, we evaluated the pharmacokinetic of the tablet in comparison with the mixture (Mix-160). For this, a prevalidated and optimized analytical methodology was used by a LC-MS/MS system for quantifying analytes in rodent plasma samples [8]. Our results allowed us to establish the pharmacokinetic basis for extrapolating in clinical trials those that assess the efficacy and safety of a citroflavonoid-based drug for the treatment of diabetes and hypertension.

## 2. Results and Discussion

### 2.1. Quality Control

Intra and interday precision and accuracy were calculated in three concentrations (2.5, 7.5, and 12.5 ng/mL). The calculated mean intraday precision was 4.11 ± 1.25%, with a sampling recovery of 92.76 ± 3.67%. Interday precision was 6.54 ± 1.59%, with a sampling recovery of 93.57 ± 1.89%. According to international guidelines, recovery intervals are between 60% to 115%, with a precision of 21% to 32% [9,10]. Results showed that previous evidence reported by Araujo-León et al. [8] showed a precision lower than 10% and an excellent recovery of more than 90% in picograms concentration order. This was compared to our results with research with flavonoids with a similar chemical structure that were evaluated in rat serum samples analyzed by LC-MS/MS. The results showed that the recovery of myricitrin was between 87.48% to 97.37% with a precision of <12.82% [11]. Another study to quantify isoquercetin and astragalin reported precision between 1.4% to 10.72% and accuracy >94.78% [12].

Figure 1A showed no significant interfering peaks in the blank sample. In addition, good chromatographic resolution (Rs ≥ 1.5) was observed in the spiked sample (Figure 1B). Matrix interferences were reduced by the use of a QqQ mass spectrometer, which is characterized as having a high specificity in multiple reaction monitoring experiments. In this case, we performed the analysis in negative mode for what we observed as a hesperidin deprotonated molecule [M − H] at *m*/*z* 609.2 as a precursor ion and a signal *m*/*z* 301.1 as product ion after ion collision. Similar results were observed with an internal standard quercetin: [M − H] at *m*/*z* 301.1 and *m*/*z* 151.1, as well as an internal standard naringenin [M − H] *m*/*z* 271.1 and *m*/*z* 151.1.

Finally, the chromatographic method showed good linearity with a correlation coefficient larger than 0.99 to each signal. LOD and LOQ of hesperidin were of 0.3 ng/mL and 1.1 ng/mL, respectively. These above values for naringenin were 0.2 ng/mL and 0.5 ng/mL. Based on previous results, LOD and LOQ for each flavonoid were highly sensitive which suggest that this method should be useful in pharmacokinetic studies. [8,11,12,13].

### 2.2. Pharmacokinetic Evaluation

Graphics of mean concentration of hesperidin and naringenin in blood samples versus time are represented in Figure 2A–D. All pharmacokinetic curves showed a first-order kinetic behavior by visual inspection. The results of the calculation by a noncompartmental model of pharmacokinetic parameters are shown in Table 1.

In the current study, we observed that the first detectable concentration of the first sample was at 5 min. These results were associated with the absorption and elimination equations; the literature described hesperidin and naringenin found between 5 and 15 min after oral administration [14,15]. Finally, concentrations lower than 3 ng/mL were detected at 24 h after administration. This effect was reported by Chen et al. [16], which they described as a enterohepatic recirculation phenomena before final elimination.

Statistical analysis was performed to compare each pharmacokinetic parameter of a single flavonoid into the mixture and the tablet dosage form. Overall, results did not show statistical differences (*p* > 0.05) in both administrations. A first observation was that the component in the tablet did not have a significant influence in the pharmacokinetic to both flavonoids. However, we observed that K_abs_ > K_e_, as shown in Figure 1 and Table 1. This suggested that the number of flavonoids in the intragastric compartment were more significant than in blood. This phenomenon was observed until T_max_, where flavonoids in the blood modify the equilibrium ratio between absorption and elimination, for which CL/F was more remarkable than absorption [17].

Both flavonoids did not show statistical differences in C_max_ (*p* > 0.05). We observed an increase of 0.67 h in T_max_ when hesperidin was added to the tablet and similar C_max_ for both preparations (mixture vs. tablet). Although there were no statistically significant differences (Figure 2A in contrast to Figure 2B), data suggest that the absorption rate slope was faster when administered as a tablet. When equilibrium is achieved in biophase, it is essential to re-quantify between 4 to 10 h post administration with more samples across this time lapse to establish the T_max_. Because hesperidin is a glycoside flavonoid, it could have hydrolysis in the gastric compartment and modify the absorption ratio. Conversely, naringenin was not similar in T_max_ and C_max_ parameters compared with hesperidin. Based on the investigation of Escudero et al. [18], glucoronate metabolites of naringenin were not found at 2 h after a single oral administration. However, concentration of these metabolites began to increase, reaching the maximum value after 6 h; at 10 h, they were undetectable. When we compared our results with those reported by Escudero et al. [18], we found an accelerated elimination of naringenin between 2 and 12 h.

To compare the pharmacokinetics results of hesperidin, we observed reports in the literature with a C_max_ 3.5–4 h after single oral administration [19], which matches with our study because the absorption elimination equilibrium was achieved at 4 to 6 h in both cases. For instance, [20] found glucuronide and sulfoglucuronide metabolites from 3 h reaching their maximum concentration between 5 and 7 h after a single oral administration of both hesperidin and narirutin. In this context, the explanation of an absorption elimination equilibrium at 4–6 h could be the intrinsic metabolism reactions. However, hesperidin had a rapid elimination due to the enzymatic biotransformation by CYP450. In addition, it could be seen that more than 95% of serum flavonoids were eliminated at 12 h post administration (Figure 2A–D).

First-pass metabolism is an essential biological process to discuss on flavonoid metabolism. For instance, [21] described that flavonoid first-pass metabolism in gastrointestinal tracts is regulated by lactase-phlorizin hydrolase and gut microbiota. It was described that naringin is hydrolyzed to naringenin and absorbed into cells by passive diffusion and active transport. This group identified 15 phenolic catabolites, such as propionic and benzoic acid derivatives. In addition, it was reported that second-pass metabolism produced a total of 23 naringin metabolites, which were mostly glucuronides and sulfate conjugates, including naringenin-7-O-sulfate, and naringenin-O-glucuronide-O-sulfate. Because of the first-pass metabolites corresponded to nonabsorbed flavonoids that were excreted by enterohepatic circulation, results suggested that all flavonoids did not reach biophase. Actually, part of the dose was eliminated in feces, which highlights the importance to study clearance (CL) and volume of distribution (VD) based on absorbed drug fraction (F). Our results showed a low concentration of both flavonoids in serum samples compared with oral dosage administered (161 mg/kg). Elimination in feces could explain the small AUCs found in our study.

In first-pass metabolism in the liver, CYP450 enzymes play an essential role in the biotransformation of the metabolites [14] compared to the dose-dependent behavior of naringin when different doses were used and different pharmacokinetic parameters calculated. This report showed that there was no linear correlation between them. They measured free naringin and total naringin (equivalent with its metabolites) showing differences in biotransformation proportion in a dose-dependent fashion. Naringenin is a hydrolysis product of naringin and reaches a higher transformation between 2 and 4 h.

Flavonoids are metabolites widely spread in the Citrus genus. It is well known that these compounds interact with many isoforms of the cytochrome complex, with CYP3A4 being the isoform most studied [22]. Naringenin and hesperidin also showed significant inhibition of CYP1A2 [23], CYP2C9, CYP2D6, CYP3A4/5, and CYP1A2 [24]. Interestingly, Luong et al. [25] showed that 100 µmol/mL naringenin inhibited its own in vitro hydroxylation. This evidence could be the cause of why naringenin was not hydroxylated when it was combined with hesperidin. Finally, CYP2C9 inhibition by hesperidin and diosmin was reported by [26] as what made their metabolism and CL/F more slower.

## 3. Materials and Methods

### 3.1. Chemical and Reagents

Methanol, acetonitrile, and water (J.T. Baker, Phillipsburg, NJ, USA) were chromatographic grade. Analytic grade formic acid (99%) and dimethyl sulfoxide (99%) were purchased from Sigma-Aldrich (St. Louis, MI, USA). Finally, analytical grade (>98%) hesperidin and naringenin for chromatographic analysis and pharmaceutical grade (>90%) for tablet manufacturing were purchased from Sigma Aldrich (St. Louis, MI, USA). Microcrystalline cellulose (Avicel^®^ PH-102), agglomerated α-lactose monohydrate (Tablettose^®^ 70), sodium croscarmellose (JRS Pharma, Patterson, NY, USA), sodium lauryl sulfate (Cosmopolitan Drug Store, Mexico City, Mexico), and magnesium stearate (Central de Drogas, S.A. de CV, Mexico City, Mexico) were purchased from local sources. Active pharmaceutical ingredients and excipients used in formulation development were analytical and pharmaceutical grade, respectively.

### 3.2. Tablet Manufacturing

Immediate release tablets (300 mg) were manufactured by direct compression in a Zhemarck hydraulic press (Zhemarck, Rome, Italy). We used a 10 mm awl and applied 1400 psi compression for 5 s. Before the compression process, a powder mixture was prepared as follows: MIX-160 was weighed in a polypropylene container, excipients were added one by one starting with sodium lauryl sulfate and magnesium stearate as surfactant and lubricant, respectively. Avicel^®^ PH-102 and Tablettose^®^ 70 (1:1 mass/mass ratio) were used as diluents for their compressibility-flowability properties and sodium croscarmellose as a disintegrant. All ingredients were mixed in a double-coned polypropylene container for 10 min until homogenization.

### 3.3. Equipment

The LC system was an Agilent Technologies model 1290 series (San Jose, CA, USA) equipped with an autosampler and a quaternary solvent manager with an online degasser. The LC system was coupled to a mass spectrometer of triple quadrupole (QqQ) from Agilent Technologies model 6470 (San Jose, CA, USA) equipped with a Jet Stream Electrospray Ionization Source. We used the same operating conditions in the ionization source: spectra collected by multiple reaction-monitoring experiments in negative-ion mode and the transition ions previously reported by Araujo-León et al. [8].

### 3.4. LC-MS/MS Quality Control

Quality control was performed according to the reports of Araujo-León et al. [8]. Validation was performed by evaluation of inter and intraday precision and accuracy, repeatability, specificity, linearity, limit of detection (LOD), and limit of quantification (LOQ). We used a spiked sample of murine serum and tested it five times by the sample preparation method as a blank.

### 3.5. Animals

Male Wistar rats around 250 g were obtained from the animal house at Universidad Júarez Autónoma de Tabasco. Animals were kept in standard laboratory conditions with a 12 h light and dark cycle at 25 °C and humidity around 65%, with certified diet and water ad libitum. All experiments met regulations of the Mexican Federal Regulations for Animal Experimentation and Care (NOM-062-ZOO-1999) and the 2010 Guide for the Care and Use of Laboratory Animals (No. 85-23, revised edition). An ethics committee approved this animal experimentation protocol in October 2017 with the certification code 2017-001.

### 3.6. Sample Preparation

Chemical extraction of naringenin and hesperidin was performed by solid phase extraction with C18 cartridges as reported by Araujo-León et al. [8]. Finally, samples were collected in 2 mL amber vials and transferred for analyzing in LC-MS/MS equipment. All the samples were immediately analyzed after extraction at 4 °C.

### 3.7. Pharmacokinetic Evaluation

Experimental animals were divided into two groups: (1) rodents treated with Mix-160 (pure flavonoids) and (2) rodents treated with the tablet (flavonoids + excipients). Oral administration in both groups was carried out by intragastric gavage at a single dose of 161 mg/kg of Mix-160. We collected a blood microsample (0.5 mL) from the tail vein between 5 min to 1440 min after oral administration (specific times and conditions were reported by Araujo-León et al. [8]. Blood samples were transferred to a 2 mL centrifuge tube to centrifugate at 3000 rpm for 10 min. Finally, the serum sample was collected and stored at −80 °C until the quantification analysis. 

### 3.8. Software and Statistical Analysis

The pharmacokinetic parameters shown in Table 1 were evaluated by a noncompartmental model. Each sample was tested in triplicate. The pharmacokinetic parameters were analyzed in WiNolin 2.0 software (Certara USA, Inc., Princeton, NJ, USA) regarding a noncompartmental model. Statistical analysis was performed in SPSS 20.0 (IBM Corp., Armonk, NY, USA) software. Statistical significance was achieved when *p* < 0.05. Results of flavonoid quantification were expressed as mean ± standard deviation.

## 4. Conclusions

Both presentations of naringenin/hesperidin (Mix-160 vs. tablet) did not show statistical differences (*p* > 0.05) of pharmacokinetic parameters in a first-order kinetic model of absorption and a noncompartmental model of elimination. Both flavonoids showed a latency time of zero which means that the initial absorption began immediately after intragastric gavage. The first time, we reported that initial concentration of both flavonoids was detectable at 5 min after intragastric administration. In addition, we reported a concentration lower than 3 ng/mL at 24 h, which suggested enterohepatic recirculation phenomena before final elimination. In addition, this evidence suggests a high half-life in the organism possibly associated with enzymatic activity of CYP450 in the liver and the first-pass metabolism.

We concluded that the tablet dosage form did not modify the pharmacokinetics of both flavonoids. These results and our previous findings [4] point to Mix-160 in tablet dosage form as being a feasible and effective candidate for developing drugs in the treatment of diabetes and hypertension.

## Figures and Tables

**Figure 1 molecules-27-00391-f001:**
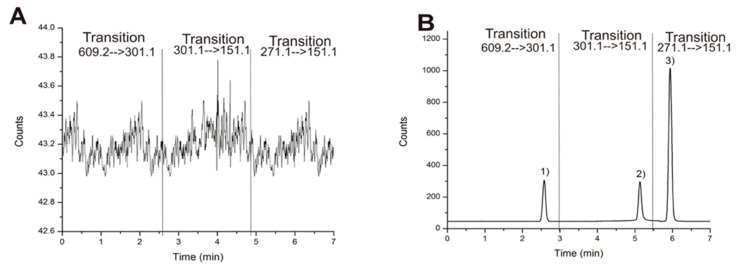
(**A**) Blank plasma matrix chromatogram and (**B**) spiked sample chromatogram at 10 ng/mL b calculated by multiple reaction monitoring with each transition. (1) hesperidin, (2) quercetin, and (3) naringenin.

**Figure 2 molecules-27-00391-f002:**
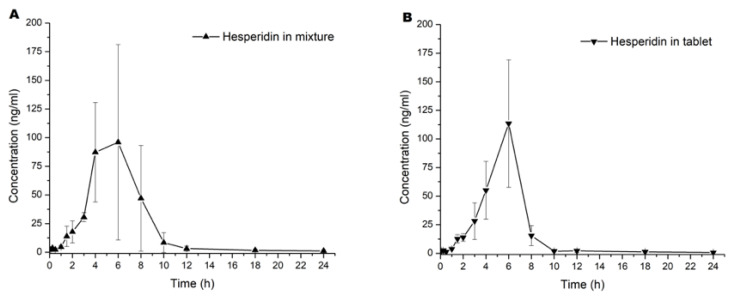
Pharmacokinetic curves of the serum mean concentration versus time of (**A**) hesperidin in mixture, (**B**) hesperidin in tablet, (**C**) naringenin in mixture, and (**D**) naringenin in tablet. Values are expressed in mean ± SD.

**Table 1 molecules-27-00391-t001:** Pharmacokinetic parameter results of hesperidin and naringenin in Mix-160 and tablet.

Parameter	Mean ± SD			
	Hesperidin in Mix-160	Hesperidin in Tablet	Naringenin in Mix-160	Naringenin in Tablet
k_abs_	0.39 ± 0.02	0.45 ± 0.04	0.79 ± 0.06	0.64 ± 0.03
k_e_	0.24 ± 0.05	0.25 ± 0.03	0.40 ± 0.02	0.41 ± 0.02
T _½ elim_ (h)	7.71 ± 1.55	8.67 ± 1.15	4.67 ± 0.61	4.02 ± 0.65
T_max_ (h)	4.66 ± 1.15	5.33 ± 1.15	2.00 ± 00	2.00 ± 00
C_max_ (ng/mL)	120.00 ± 72.46	121.352 ± 43.70	472.308 ± 92.18	515.937 ± 142.78
AUC _0→24_ (ng/mL·h)	516.10 ± 146.10	410.80 ± 85.58	1538.14 ± 124.40	1648.57 ± 181.30
AUC_0→∞_ (ng/mL·h)	522.12 ± 147.27	413.79 ± 86.14	1543.64 ± 191.9	1653.14 ± 181.70
Vd/F (L/kg)	745.17 ± 558.31	732.78 ± 200.98	151.93 ± 29.2	141.79 ± 44.25
CL/F (L/kg·h)	157.96 ± 78.62	173.14 ± 38.44	60.20 ± 7.32	56.614 ± 11.24
MRT (h)	6.51 ± 1.08	6.22219 ± 0.4776	4.38102 ± 0.134	4.32512 ± 0.38

k_abs_: absorption constant, k_e_: elimination constant, T_½_ elim: elimination half-life, T_max_: time of maximum concentration, C_max_: maximum concentration, AUC_0→24_: area under the concentration–time curve from 0 to 24 h, AUC_0→∞_: area under the concentration–time curve from 0 to infinite, Vd/F: apparent volume of distribution, CL/F: oral clearance, MRT: mean residence time, SD: standard deviation.

## Data Availability

All relevant data are included in the article.

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
