# Peer review of "A Pharmacokinetic Study of Mix-160 by LC-MS/MS: Oral Bioavailability of a Dosage Form of Citroflavonoids Mixture"

_molecules, 2022, doi:10.3390/molecules27020391_

Round 1

Reviewer 1 Report

The manuscript entitled “A Pharmacokinetic Study of Mix-160 by LC-MS/MS: Oral bioavailability of a dosage form of citroflavonoids mixture” by Araujo-León et al., apply an optimized LC-MS/MS protocol published by the authors in Molecules 2020, 25, 4241., to evaluate the pharmacokinetics parameters of two different formulations of hesperidin and naringenin. Considering the potential of these citroflavonoids as natural-based drugs, the results presented are interesting, and suitable to published after minor corrections in the mass spectrometry section.

Comments:

Lines 87-90

“a hesperidin adduct (M-1)” should be changed  to  “a hesperidin deprotonated molecule [M-H]-

m/z abbreviation is written in italic (Definitions of terms relating to mass spectrometry.  IUPAC recommendations 2013) 

The signal = between  m/z and the value must be removed (m/z 301.1 is the corrected form)

Please corrected 15.1 to 151.1 .

Author Response

Response to Reviewer 1 Comments

Thank you so much for your comments, all of each were very really enriching for our paper, below we attend each of one.

Point 1: “a hesperidin adduct (M-1)” should be changed to “a hesperidin deprotonated molecule [M-H]-

Response 1: We change in the sentence to “a hesperidin deprotonated molecule [M-H]-

Point 2: m/z abbreviation is written in italic (Definitions of terms relating to mass spectrometry.  IUPAC recommendations 2013) 

Response 2: We change all to italic when we cited m/z in the manuscript.

Point 3: The signal = between  m/z and the value must be removed (m/z 301.1 is the corrected form)

Response 3: We delete all the = signals in the manuscript that was between m/z.

Point 4: Please corrected 15.1 to 151.1 .

Response 4: Done

Reviewer 2 Report

The manuscript deal with a pharmacokinetic study on a particular citrus flavonoid mixture (Mix-160) by LC-MS/MS. The whole study is properly constructed and executed. The results were clearly represented by means of figures and tables that were thoroughly discussed and correlated with existing literature data. The manuscript is suitable for publication in its present form. On page 2, line 87: "a hesperidin adduct" is not correct and should be replaced with "a hesperidin deprotonated molecule".

Author Response

Response to Reviewer 2 Comments

Thank you so much for your comments, all of each were very really enriching for our paper, below we attend each of one.

Point 1: The manuscript deal with a pharmacokinetic study on a particular citrus flavonoid mixture (Mix-160) by LC-MS/MS. The whole study is properly constructed and executed. The results were clearly represented by means of figures and tables that were thoroughly discussed and correlated with existing literature data. The manuscript is suitable for publication in its present form. On page 2, line 87: "a hesperidin adduct" is not correct and should be replaced with "a hesperidin deprotonated molecule".

Response 1: We change in the sentence to “a hesperidin deprotonated molecule [M-H]-

Reviewer 3 Report

The authors describe a PK study for determining the PK parameters of a mixture of hesperidin and naringenin.

The paper needs an extensive revision of english language and a revision of validation parameters. The reviewer suggests that EMA or FDA guidelines should be adopted for the verification of the method's performance.

Author Response

Response to Reviewer 3 Comments

Thank you so much for your comments, all of each were very really enriching for our paper, below we attend each of one.

Point 1: The authors describe a PK study for determining the PK parameters of a mixture of hesperidin and naringenin. The paper needs an extensive revision of the English language and a revision of validation parameters. The reviewer suggests that EMA or FDA guidelines should be adopted for the verification of the method's performance.

Response 1: Dear Reviewer 3, thank you so much for your comments. In this manuscript, we apply an optimized and validated method according to the ICH Q2A by LC-MS/MS published previously in this journal (Araujo-León et al 2020, https://doi.org/10.3390/molecules25184241), in this paper you can see the full validation covering the next parameters: specificity, Linearity, Limited of Detection (LOD), and Limited of Quantification (LOQ), Precision and Accuracy, Recovery and Matrix effect.

In this manuscript, in section 3.4 LC-MS/MS quality control, we cited the previous paper, and clarify that only using the validation was performed by evaluation of inter and intra-day precision and accuracy, repeatability, specificity, linearity, the limit of detection (LOD), and limit of quantification (LOQ) just as quality control because the method was fully validated previously.
